# miR-615 Fine-Tunes Growth and Development and Has a Role in Cancer and in Neural Repair

**DOI:** 10.3390/cells9071566

**Published:** 2020-06-27

**Authors:** Marisol Godínez-Rubí, Daniel Ortuño-Sahagún

**Affiliations:** 1Laboratorio de Investigación en Patología, Departamento de Microbiología y Patología, CUCS, Universidad de Guadalajara, Guadalajara, Jalisco 44340, Mexico; 2Laboratorio de Neuroinmunobiología Molecular, Instituto de Investigación en Ciencias Biomédicas (IICB) CUCS, Universidad de Guadalajara, Guadalajara, Jalisco 44340, Mexico

**Keywords:** microRNAs, cancer, miR-615, miR-615-5p, miR-615-3p, cell growth, cell differentiation, tumor suppressor, tumor promoter, neural repair, oncogene

## Abstract

MicroRNAs (miRNAs) are small noncoding RNAs that function as epigenetic modulators regulating almost any gene expression. Similarly, other noncoding RNAs, as well as epigenetic modifications, can regulate miRNAs. This reciprocal interaction forms a miRNA-epigenetic feedback loop, the deregulation of which affects physiological processes and contributes to a great diversity of diseases. In the present review, we focus on miR-615, a miRNA highly conserved across eutherian mammals. It is involved not only during embryogenesis in the regulation of growth and development, for instance during osteogenesis and angiogenesis, but also in the regulation of cell growth and the proliferation and migration of cells, acting as a tumor suppressor or tumor promoter. It therefore serves as a biomarker for several types of cancer, and recently has also been found to be involved in reparative processes and neural repair. In addition, we present the pleiad of functions in which miR-615 is involved, as well as their multiple target genes and the multiple regulatory molecules involved in its own expression. We do this by introducing in a comprehensible way the reported knowledge of their actions and interactions and proposing an integral view of its regulatory mechanisms.

## 1. Introduction

Homeobox genes encode transcription factors that regulate the expression of several genes during embryogenesis [1]. Among them, Hox genes are very well described as specifying the anteroposterior pattern as well as the development of many organs [2], for example during neurodevelopment as well as in malignant glial tumors and as gliomas and glioblastomas [3]. Within this gene cluster, introns represent a hypothetically rich source of microRNAs (miRNAs). miRNAs are single stranded noncoding 18–25 nucleotide RNAs that post-transcriptionally attenuate gene expression [4]. Among them is miR-615, which is located within the intron of the Hoxc5 gene (12q13.13). miR-615 has a restricted phylogenetic distribution and is absent in non-mammalian tetrapods but highly conserved across eutherian mammals [5], which allows it to contribute in eutherian evolution and development [6].

While miR-615 expression can be coupled to HOXC5 expression, it can also be transcribed independently using an intragenic promoter [6]. A human miR-615 sequence obtained from miRbase is 96 nucleotides long (Figure 1A) and constitutes a highly conservative sequence, which can be seen in the consensus sequence obtained from several vertebrates (Figure 1B). These contain two different segments (miR-615-3p and miR-615-5p of 21 nucleotides each) which can perform interference activities and function as regulatory miRNAs. This sequence forms a hairpin or stem-loop (Figure 1C) where the two sides of the stem contain the corresponding sequences miR-615-3p and miR-615-5p [7].

## 2. Functions of miR-615

At present, more is known about the involvement of miRNA-615 in various pathologies than about its physiological function. miR-615 is highly expressed in the mouse embryo and has therefore been involved during embryogenesis in the regulation of growth and development, as suggested by target prediction and transcriptomic analyses. After analyzing miR-615 expression in several human cell lines and tissues, only very few cell lines lacked its expression [6]. The gonads, the kidney and the cerebellum are among the principal tissues in various species in which miR-615 expression occurs.

During embryonic development in rats, the expression of miR-615-3p was detected from the fourteenth day and was significantly reduced during osteogenic and adipogenic differentiation in a time-dependent manner. In bone marrow mesenchymal stem cells (BMSCs) induced with chondrogenic differentiation, the mRNA expression of specific related genes COL2A1, COL10A1, ACAN and MATN3 was significantly decreased following miR-615-3p overexpression. In addition, the expression of SOX9, a transcription factor promoting chondrogenesis, was also significantly decreased, while knockdown of miR-615-3p obtained the opposite result. This indicates that miR-615-3p inhibits the expression of chondrogenic-specific genes and may inhibit chondrogenic differentiation [8].

On the other hand, some studies on cancer cell lines have involved this miRNA in the regulation of cell growth, proliferation and migration [9], with it acting as a tumor suppressor [6] or as a tumor promoter [10,11,12].

## 3. miR-615 Targets Under Physiological and Pathological Conditions

Although the same miRNA can regulate several genes, it can also be regulated by several genes, and one gene can be targeted by multiple miRNAs. Therefore, it is very important to be able to identify the specific targets of each miRNA in order to understand their participation in different pathologies and identify new therapeutic targets. miR-615-3p has mainly been involved in the inhibition of cell differentiation during osteogenesis, chondrogenesis and alveolar epithelial cells, in addition to its participation in the cirrhotic process derived from hepatitis both in the liver and in the spleen through targeting different genes (Table 1).

The involvement of miR-615-3p during osteoblastogenesis and skeletal development has been reported [15], as its expression has been deregulated during glucocorticoid-mediated bone loss. It has been determined that overexpression of miR-615-3p significantly reduces the expression of the FOXO1 and GDF5 proteins, two genes related to osteogenesis, while inhibition of miR-615-3p increases their expression. Therefore, miR-615-3p negatively regulates osteogenic differentiation by post-transcriptional suppression of osteogenic regulators.

It is interesting that miR-615-3p also promotes osteoarthritis (OA), most probably by inhibiting the chondrogenic differentiation of bone marrow mesenchymal stem cells (BMSCs). Overexpression of miR-615-3p significantly reduced SOX9 expression and increased the expression of proinflammatory cytokines like IL-1, IL-6, IL-8 and TNF-α, which were significantly reduced following miR-615-3p knockdown. Since SOX9 could regulate chondrogenic differentiation, miR-615-3p promotes the expression of inflammatory cytokines by inhibiting the chondrogenic differentiation of BMSCs [8].

miR-615-3p and inflammatory factors (IL-1, IL-6, IL-8 and TNF-α) are overexpressed in peripheral blood of patients with neonatal acute respiratory distress syndrome (ARDS). Therefore, it has been suggested that miR-615-3p participates in the progression of ARDS by inhibiting the differentiation of BMSCs to alveolar type II epithelial cells (ATII) through inhibiting the Wnt/β-catenin pathway [16], given that under some circumstances, BMSCs can differentiate to ATII [25].

miR-615-3p has been noted to regulate hypersplenism (portal hypertension related to cirrhosis) of splenic macrophages in cirrhosis related to the hepatitis B virus [26]. In addition, it improves their phagocytic ability by acting on the ligand-dependent nuclear receptor corepressor (LCoR), which in turn may suppress the peroxisome proliferator-activated gamma receptor (PPARγ) [13]. Therefore, an overexpression of miR-615-3p decreases LCoR expression, which reduces the expression of PPARγ, decreasing the phagocytic capacity of splenic macrophages.

In nonalcoholic fatty liver disease (NAFLD), palmitate, a lipotoxic saturated free fatty acid, induces apoptosis and requires maximal expression of CHOP (C/EBP homologous protein), a proapoptotic transcription factor. This is achieved by inducing endoplasmic reticulum stress and a concomitant downregulation of repressive miR-615-3p, promoting hepatocyte lipoapoptosis. Thus, the reduction of miR-615-3p by palmitate-induced ER stress derepresses the expression of one of its targets, the CHOP protein, and defines its sensitivity to cell death without contributing to its basal regulation [14].

On the other hand, the involvement of miR-615-5p in the reparative processes and angiogenesis has been demonstrated in various contexts. In human liver tissue, miR-615-5p is found to be moderately upregulated in cirrhotic liver [27], as well as in plasma from patients with different degrees of liver fibrosis [28]. Therefore, the authors suggested that it may be relevant as a biomarker in the liver repair process.

In patients with diseases associated with vascular dysfunction, such as acute coronary syndrome (ACS), diabetes mellitus (DM) or systemic arterial hypertension (SAH), miR-615-5p is overexpressed in peripheral blood [29,30]. In experimental models of vascular dysfunction and tissue damage, including wounds in diabetic mice and human skin organoids, miR-615-5p significantly inhibits VEGF, AKT and eNOS signaling, which could suppress the angiogenic signals and restorative processes such as granulation tissue formation and wound healing. This effect is achieved by directly targeting IGF2 (insulin-like growth factor 2) and RASSF2 (Ras association domain family member 2), as an accumulative effect [29]. The IGF-axis is also modulated by miR-615-5p in NK cells of hepatocellular carcinoma (HCC) patients, in which upregulation of miR-615-5p directly represses IGFR1 signaling and reduces the cytotoxic markers NKG2D, TNF-a and perforins. This triggers an anti-cytotoxic effect in NK cells and potentially attenuates their anti-tumor activity [31].

Ischemic retinopathies like diabetic retinopathy and oxygen-induced retinopathy of prematurity are characterized by vascular dysfunction and pathological angiogenesis with capillary degeneration, altered permeability and inflammation [32]. In human umbilical vein endothelial cells (HUVEC) and mouse models of retinal vasculature damage, overexpression of miR-615-5p partially rescues endothelial cells (ECs) from hyperglycemic and hypoxic stress-induced cell apoptosis, suppresses pathological angiogenesis and increases the viability, cell migration and tube formation of ECs. These effects are mediated by the inhibition of Myocyte-specific enhancer factor 2A (MEF2A), Tyrosine-protein kinase receptor Tie-2 (Tie2) and IGF2 [33], all three of which are involved in proliferation and the angiogenesis process [34,35,36]. These observations show clinical correlation in the retinas and plasma of diabetic patients, as well as in patients with ACS and SAH. Therefore, modulation of miR-615-5p signaling may be considered a potential therapeutic target in diseases with defects in angiogenesis such as diabetic retinopathy, coronary artery disease or peripheral vasculopathy. On the other hand, the anti-angiogenic effect of miR-615-5p has been documented as part of its ability to suppress tumor growth (see below).

Therefore, miR-615 has been related to the inhibition of cell differentiation during osteogenic or chondrogenic processes, and to immune, inflammatory, and reparative responses in macrophages, splenocytes, hepatocytes, BMSCs and endothelial cells. However, it also participates in the development or suppression of many tumors.

## 4. mir-615-3p Dual Role as Oncogene and Tumor Suppressor in Cancer

miRNAs act primarily by regulating gene expression by binding to the 3-UTR of the target mRNA. Derived from this mechanism of action, they can participate both as tumor promoters and suppressors in the development of various types of cancer [37,38,39] (Table 1).

miR-615-3p has been identified as a tumor suppressor in cancers including lung [18], esophageal [22] and renal [19]. In contrast, overexpression of miR-615-3p is involved in other cancers, such as hepatocellular cancer [10], gastric cancer [11] and prostate cancer [12].

In non-small cell lung cancer (NSCLC), decreased expression of miRNA-615-3p has been described [40]. In vitro overexpression of this miRNA significantly inhibits cell proliferation and migration, as well as tumor growth and metastasis. This is mediated by the binding of miRNA-615-3p to the 3’-UTR insulin-like growth factor 2 (IGF2) [18]. In addition, overexpression of IGF2 has been shown to rescue the inhibitory effect of miR-615-3p on cell proliferation, migration and invasion, confirming that it is a direct functional target in NSCLC [18].

In most human somatic cells, when cell differentiation is induced, hTERT (a subunit of telomerase) is suppressed and eventually silenced [41,42]. However, increased telomerase expression has been observed in more than 85% of cancers [43], which can be dramatically reduced by overexpression of HOXC5 and miR-615-3p. This is the case both in human cancer cell lines and during differentiation of pluripotent cells [20], in which the expression of HOXC5 and miR-615-3p is suppressed but significantly activated during cell differentiation [44,45] compared to the expression of hTERT in differentiated cells. Both can suppress hTERT through an upstream enhancer region (transcriptional pathway) and 3′UTR (post-transcriptional pathway), respectively, forming a feed loop to negatively regulate hTERT mRNA expression, telomerase activity and telomere elongation. The key role in hTERT suppression, however, is played by HoxC5, while miR-615-3p plays a secondary role in the negative regulation of hTERT by binding to its 3′UTR [20].

The first evidence of the oncogenic functions of miR-615-3p came from findings in malignant mesotheliomas [46]. Subsequently, its overexpression in poorly differentiated colorectal cancer (CRC) has been detected [47] in a differentially regulated manner between the right and left colon, in both normal and cancerous tissue [48], and it has been proposed as a biomarker of prognosis [49].

The expression of miR-615-3p in hepatocellular carcinoma (HCC), on the other hand, is higher in the recurrence group than in the non-recurrence group, where it increases proliferative activity and promotes invasiveness. Additionally, miR-615-3p has been associated with HCC recurrence by inducing chemoresistance and the acquisition of the epithelial-mesenchymal transition (EMT) [10]. Overexpression of miR-615-3p has also been described as leading to a poor prognosis in metastatic kidney cancer [50]. These studies do not, however, reveal a mechanism or target gene for these actions.

The oncogenic function of miR-615-3p in gastric cancer has been reported recently [11]. In gastric cancer cells (cell line SGC7901), overexpression of miR-615-3p promotes proliferation and migration by suppressing CELF2 (tumor suppressor CUGBP- and ETR-3-like family 2) expression. More recently, elevated miR-615-3p expression has also been implicated in poor prognosis of postoperative biochemical recurrence (PBR) and prostate cancer (PC) specific survival. Since it increases the viability, proliferation and migration of PC3M prostate cancer cells [12] and is associated with a worse prognosis in PC, miR-615-3p is indicated as an oncogenic driver of this cancer.

Moreover, in breast cancer tissues and breast cancer cell lines, miR-615-3p level is also upregulated, and promotes metastatic ability by targeting 3’-UTR of PICK1, inhibiting it, and thus increasing TGF-β signaling. By contrast, PICK1 expression exerts the opposite effect, acting as a negative feedback loop for TGF-β signaling by inhibiting the binding of DICER1 to Smad2/3 and the processing of pre-miR-615-3p to mature miR-615-3p [17].

In summary, while the role miR-615-3p plays in different types of cancers appears to be somewhat contradictory and misleading, it may simply be indicative of its function in a highly cell- and disease-type-specific manner. For this reason, it is imperative that there be further investigation of miR-615-3p’s targets (Table 1), which will be critical in accurately understanding the role of miR-615-3p in cancer.

## 5. miR-615-5p as a Tumor Suppressor in Several Types of Cancer

Cumulative evidence indicates that miR-615-5p functions as a tumor suppressor through a post-transcriptional inhibition of oncogenes involved in essential biological processes. It does so mainly by preventing invasiveness and metastasis, the main causes of tumor reappearance. With the exception of some examples of HCC [27,51], a downregulation of this miRNA has been documented in several malignancies of different lineages such as carcinomas [28,52,53], lymphomas [54] and glioblastomas [55], with an indirect relationship between the level of expression of miR-615-5p and the tumor cell capacity for proliferation, invasion, migration and angiogenesis. Their target genes are summarized in Table 2 and Figure 2.

Similarly to its 3p counterpart, miR-615-5p suppresses the transcription of numerous oncogenes through specific binding to sequences in the 3’-UTR region, such as AKT [9,29,56,59,60], CCND2 [62] and IGFR1/IGF2-axis [27,31,56,57,58,64]. In pancreatic adenocarcinoma (PAC), miR-615-5p has reduced expression in malignant cells compared to adjacent normal pancreatic acinar cells. The overexpression of miR-615-5p in PAC murine models is associated with a significant reduction in tumor growth rate, weight and volume compared to negative controls. Similarly, it alleviates the rate of proliferation and the metastatic potential while promoting tumor cell apoptosis. These effects are carried out through negative regulation of AKT2 by miR-615-5p [9,59]. This mechanism of gene expression silencing also applies to AKT1, AKT2, IGF2 and SHMT2 in NSCLC tissues [56,60]. In the case of IGF2, it has also been demonstrated in ESCC [58], PAC [57] and HCC tissues [27,64]. The inhibitory effect on AKT is not limited to the repression of translation, but also to phosphorylation [9], although the mechanism of this effect has not been elucidated.

miR-615-5p expression was also markedly lower than normal cells and tissue in prostatic adenocarcinoma, in which *CCND2* mRNA (cyclin D2) was upregulated in the absence of the inhibitory effect of miR-615-5p, promoting cell cycle progression [62].

Additionally, multiple receptor tyrosine kinase (RTK) have proven to be molecular targets of miR-615-5p. This is also the case in epidermal growth factor receptor (EGFR), which is overexpressed in human tissues and cell cultures of glioblastoma multiforme (GBM). In this setting, there is a negative correlation between miR-615-5p and EGFR, so the overexpression of miR-615-5p induces the repression of EGFR and its downstream effects on growth, proliferation, invasion and migration of the tumor cells [55]. IGFR1, the IGF2-receptor, is also an RTK and a target of miR-615-5p [31]. In metastatic HCC tissues and cell lines [64] and in non-metastatic conditions [31], the IGF2/IGFR1/mTOR axis is overexpressed, which in turn correlates to the low level of miR-615-5p. The low expression was also recently documented in plasma from HCC patients [28]. This evidence is consistent with its negative role in cell proliferation, invasiveness and migration [31]. By contrast, other studies conducted on HCC tissues have documented an increase in miR-615-5p compared to control tissues [27,51]. However, the tumor suppressor effect related with its overexpression is preserved, while establishing a negative association with metastases and TNM stage [51]. In parallel, knockdown of miR-615-5p remarkably derepresses the proliferation and migration in human HCC cell lines, while overexpression reverses these effects [27,51]. The mechanism is achieved by direct and negative inhibition of miR-615-5p on its downstream targets IGF2 [27] and SHMT2 (Serine hydroxymethyltransferase 2) [51]. This enzyme participates in cell metabolism by converting serine to glycine, and its suppression has been associated with blocking growth and migration of neoplastic cells [65].

## 6. Competing Endogenous RNAs, Like lncRNAs and circRNAs, Act as a Regulatory Sponge of miR-615

It is now recognized that the noncoding portion of the genome plays an important role in a wide variety of physiological and pathological processes [66]. Various types of microRNAs compete with mRNA transcripts to form an expression-regulating mechanism [67]. These RNAs have been called competing endogenous RNAs (ceRNAs) and include lncRNAs and circRNAs that crosstalk with mRNAs and their respective miRNAs and have the ability to regulate gene expression by sequestering and competing with miRNAs [68,69] to protect mRNAs from interference. In this way, they act as molecular sponges and modulate the repressor effect of miRNAs on their target. The regulatory molecules of miR-615 are summarized in Table 3.

In the case of miR-615-3p, a number of lncRNAs were very recently discovered with the capacity to modulate its actions by sponging it. Firstly, this is the case for a HOXA transcript at the distal tip (HOTTIP). This regulates cell growth, differentiation, apoptosis and cancer progression by directly binding to miR-615-3p and acting effectively as an endogenous sponge to modulate the suppression of IGF-2 in renal carcinoma [19] or the expression of high mobility group box 3 (HMGB3) in NSCLC cells [21]. The HOTTIP oncogenic functions are mediated through negative regulation by a reciprocal repression of miR-615-3p which in turn regulates IGF-2 expression, with this molecule being a direct target gene of HOTTIP and of miR-615-3p, at least in renal cell carcinoma [19]. In NSCLC cells, on the other hand, HOTTIP acts as a molecular sponge by sequestering miR-615-3p and later regulating HMGB3, itself a direct target of miR-615-3p which promotes hypoxia-induced glycolysis. By restoring the expression of miR-615-3p or reducing the expression of HOTTIP, hypoxia-induced glycolysis is suppressed. This happens because hypoxia increases the expression of HOTTIP and suppresses the level of miR-615-3p by targeting the miR-615-3p/HMGB3 axis in NSCLC cells [21].

Another lncRNA that is an upstream regulator for miR-615-3p is LINC00657 (NORAD by noncoding RNA activated by DNA damage), that can be upregulated following DNA damage [70]. An oncogenic function for LINC00657 has been reported in several cancers such as breast [71], colorectal [72] and hepatocellular carcinoma [73]. In squamous cell carcinoma (SCC) of the esophagus cells, LINC00657 significantly increased after irradiation treatment and miR-615-3p was able to suppress proliferation and migration. In addition, LINC00657 increases the expression of JunB by suppressing the expression of miR-615-3p, and the decrease of LINC00657 may inhibit the invasion, migration and viability of ESCC cells. By analyzing the miR-615-3p target genes in common with the expression genes regulated by LINC00657 decline, JunB, a subunit of AP-1 transcription factor, was identified [22].

In non-small cell lung carcinoma (NSCLC), miR-615-5p expression is downregulated because of the decoy effect exerted by different ceRNAs, namely circ-CAMK2A [51], circRNA 100146 [61], LINC00324 [60] and lncRNA Gm15290 [56]. The overexpression of these ceRNAs (in tumor tissues and cancer cell lines) leads to a potent inhibition of miR-615-5p, while the suppression of ceRNAs produces the opposite effect. Positive regulation of the sponge molecules increases the expression of miR-615-5p target genes such as IGF2, AKT1/2 and SHMT2 [56,60], which nullifies the tumor suppressor effect of miR-615-5p and leads to cell proliferation, invasion, migration and inhibition of apoptosis. More recently, Du et al. (2019), documented the inhibition of miR-615-5p in lung adenocarcinoma through a circular RNA (cir-CAMK2A) with up-regulation of fibronectin-1. This protein is an extracellular matrix protein (ECM) and a target of miR-615-5p, which in turn promotes the activation of MMP-2 and MMP-9 matrix metalloproteases, whose activation has been associated with lymph node metastasis, distant metastasis, advanced clinical stage and poor prognosis [52]. A similar mechanism mediated by circular RNA 100146 was also described in NSCC cell lines and tissues. By capturing miR-615-5p and 3p, overregulation of multiple downstream target genes of miR-615 such as SF3B3, NFAT5, COL1A1 and MEF2C is induced. These proteins are related to the modulation of gene expression [74], cellular immune response [75], ECM proteins [76] and differentiation [77], respectively. All of them are involved in fundamental processes of tumor biology [61].

Another model employed to demonstrate the interaction between CeRNAS and miRNAS is ovarian cancer. In these patients, circPUM1 expression was positively related with poor prognosis. In vitro, overregulation of this circRNA produces sponging inhibition of miR-615-5p, which in turn derepresses the expression of miR-615 target genes such as NF-kB2 [53]. This leads to the expression of a wide range of genes related to inflammation but also to survival, angiogenesis and repair, such as MMP2, IL-8 and VEGF [78]. Thus, the sponging of miR-615-5p generates a tumorigenic phenotype. Finally, in a T-lymphoblastic lymphoma model, circ-LAMP1 (a circular RNA) produces a similar tumorigenic effect. In this model, the overexpression of circ-LAMP1 leads to the specific capture of miR-615-5p and therefore to the disinhibition of DDR2 (miR-615-5p target) [54]. Discoidin domain receptor tyrosine kinase 2 (DDR2) is an RTK related to a tumor growth promoting effect, similar to that described for the other ceRNAS [54]. The inhibitory ceRNAs of miR-615-5p are summarized in Table 3 and Figure 3.

## 7. Other miR-615-5p Repressor Mechanisms Which Contribute to the Promotion of Tumor Growth

As previously mentioned, miR-615 can be transcribed independently by regulation of its intragenic promoter [6]. It has been documented that this regulation can be influenced by transcription factors or by epigenetic mechanisms (Figure 3). CDX2 (caudal-type homeobox 2) is a transcription factor that has a role in embryonic development [79]. It is useful as a marker for carcinomas originating in the digestive tract [80], where it could act as a tumor suppressor [81]. Among its downstream targets are the Hox genes. Jiang et al. demonstrated under different pancreatic adenocarcinoma paradigms, that CDX2 binds specifically to the miR-615-5p promoter site, inhibiting its expression. In this tumor, negative modulation of the CDX2/miR-615-5p/IGF2 axis was associated with increased tumor size and weight in animal models, as well as increased expression of tumor progression markers such as Ki-67, cyclin D1, c-MYC and Bcl-2 [57]. The same suppressive effect was documented in metastatic HCC cell lines, through the modulation of the PU.1/miR-615-5p/IGF2 axis [64]. In this case, PU.1, a transcription factor with documented tumor suppressor activity in hematopoietic and lymphoid neoplasms [82,83], was shown to exert a negative regulation on miR-615-5p, which in turn was related to the invasive capacity of the neoplastic cells [64].

Finally, it has been suggested that there are epigenetic mechanisms that repress the expression of miR-615-5p through the hypermethylation of its promoter. The first such evidence was presented by Gao et al. who demonstrated that the CpG islands in the miR-615-5p promoter region are intensely hypermethylated in a tumor-dependent manner. Such repression generates a loss of inhibitory effects on IGF2 and JUNB, which in turn contributes to tumorigenesis, invasion and migration [9]. More recently, Chen et al. confirmed epigenetic silencing of miR-615-5p in an HCC model by correlating hypermethylation of the miR-615-5p promoter with downregulation of KDM4B [63], a histone demethylase, induced by p53 in response to DNA damage [84]. Secondary repression of miR-615-5p derepresses RAB24 (Ras-Related Protein 24), which in turn promotes EMT regulation by influencing the action of proteins related to cell migration and adhesion such as β1-integrin. The modulation of the axis KDM4B/miR-615-5p/RAB4/β1-integrin induced a greater capacity to migrate, proliferate, invade and generate angiogenesis in neoplastic cells [63].

## 8. miR-615 Involvement in Neural Plasticity

It worth noting that although miR-615 expression in brain tissue during embryo development is comparable to several other tissues previously mentioned in this review (consulted in mESAdb form [85]), any information regarding a possible role for this miRNA in the central nervous system (CNS) is extremely scarce.

One initial piece of evidence comes from the analysis of miR-615-3p expression in brain ischemic injuries. Following hypoxic damage in the peri-infarct region of mouse brains, miR-615-3p was downregulated in the cerebral cortex. In addition, miR-615-3p could regulate YWHAG (or 14-3-3γ), which in turn interacted with cPKCβII, γ and nPKCε-interacting protein isoforms, involved in hypoxic preconditioning induced neuroprotection [23]. Interestingly, miR-615-3p is one of five Hox cluster miRNAs significantly upregulated in the prefrontal cortex in patients with Huntington’s Disease (HD) and is related to age at death, with practically zero expression in control brains [86]. It is also one of several miRNAs with altered expression in the cerebral cortex of patients with Alzheimer’s disease (AD) [87]. Additionally, results have shown that miR-615-3p, HOXC5, PBX4, MEIS1 and MEIS2 are upregulated during the differentiation of human embryonic stem cells into neural precursors [20].

A second indication involves two works demonstrating that miR-615-5p acts as a tumor suppressor or protects against neurodegeneration in CNS related cells. Firstly, in glioblastoma (GBM) it targets EGFR expression [55], making this miRNA a novel biomarker for the early diagnosis of GBM. Intriguingly, miR-615-5p is one of multiple miRNAs upregulated after retinoic acid treatment in vitro during neural differentiation of neuroblastoma cells [88]. Secondly, miR-615-5p has also been involved in the regulation of retinal neurodegeneration, where cZNF609 acted as sponge to inhibit miR-615 activity, leading to increased METRN and rescuing its inhibitory effects on retinal glial cell proliferation [30].

A third and very recent piece of evidence comes from the demonstration that miR-615-5p directly inhibits the translation of LINGO-1 by binding to its 3’-UTR region during the differentiation of neural stem cells (NSCs) in 14-day-old brains. In this way, it facilitates neuronal differentiation in vitro and prevents the formation of astrocytes. In addition, inhibition of LINGO-1 by miR-615 promotes axonal regeneration and functional recovery in spinal cord injured rats, suggesting a possible role for miR-615 in the repair of traumatic CNS damage [24].

Taken together, this scarce but solid evidence indicates that miR-615 (both miR-615-3p and miR-615-5p) are involved in neural differentiation in the CNS during embryonic development. In adulthood, they are somehow involved in neural plasticity as a consequence of damage, such as in brain ischemia or following a spinal injury. Moreover, miR-615 seems to also be involved to a certain degree in neurodegenerative diseases such as Huntington’s [86] and Alzheimer’s [87], and most likely in others. Further research is therefore warranted.

## 9. Future Challenges

The versatility in the action of miRNAs as powerful regulators of gene expression makes them a therapeutic tool with great potential for the treatment of diverse diseases, including cancer and neurodegenerative diseases [89,90]. In this regard, the most studied strategies are: (1) the restitution of the concentration of a repressed miRNA, thus restoring its inhibitory effects on other molecules; (2) the inhibition of an overexpressed miRNA that is pathologically inhibiting some signaling pathway. However, there are still many challenges to be solved, among them, how to deliver the miRNAs to the cell of interest and how to control side effects in the context of a systemic application. Recent work has addressed this issue extensively [89,91]. So far, there is no experimental evidence of manipulation of miR-615 for therapeutic purposes.

## Figures and Tables

**Figure 1 cells-09-01566-f001:**
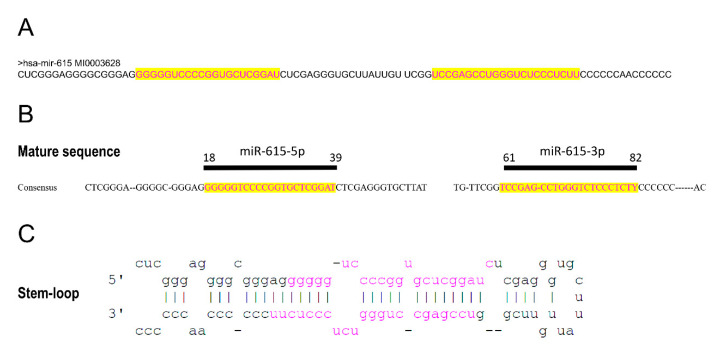
(**A**) Human miR-615 sequence obtained from miRbase. (**B**) Consensus sequence of miR-615 obtained from several vertebrates [6], including sequences of miR-615-3p and 5p. (**C**) Stem-loop of miR-615 [7].

**Figure 2 cells-09-01566-f002:**
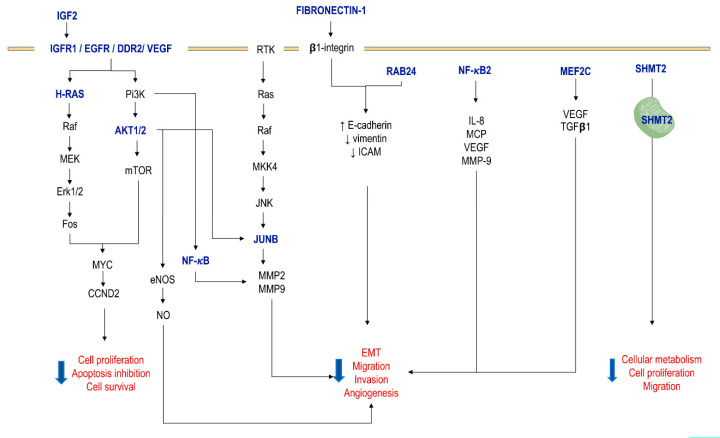
Schematic representation of the effects of miR-615-5p on cancer. The molecules that are a direct target of miR-615-5p are indicated in blue. The signaling pathways that are inhibited by the increase in miR-615-5p concentration are schematized, as well as the functional consequences that this implies, which together generate a clear tumor suppressor effect. See the text for further explanation.

**Figure 3 cells-09-01566-f003:**
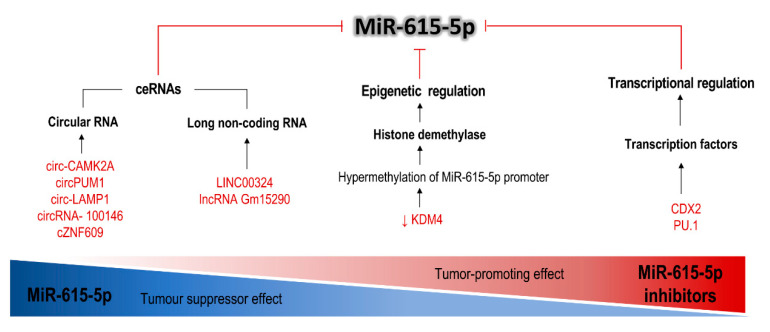
miR-615-5p can be negatively regulated under three different paradigms, according to the evidence available at present. It is possible to block the action of the molecule by ceRNAs, either circRNAs or lncRNAs. It is also susceptible to repression by hypermethylation of its intragenic promoter, which has been related to deficiencies in the action of demethylases such as KDM4. Finally, the expression of this miRNA is positively and independently regulated by transcription factors which, when repressed, cease to induce the expression of miR-615-5p. The functional consequence of this inhibition is the abolition of its tumor suppressor effect, which finally leads to the oncogenic effect associated with the decrease in miR-615-5p concentration.

**Table 1 cells-09-01566-t001:** miR-615-3p interacting genes.

Target Gene	Description	Gene Action or Effect	Cell Line	Interaction	Effect	Ref.
LCoR	Ligand-dependent nuclear receptor corepressor	Suppress PPARδ	THP-1 cells/splenic macrophages	At the 3’UTR of LCoR, located at the 193 bp downstream of the stop codon	Downregulated	[13]
CHOP	C/EBP homologous protein	A proapoptotic transcription factor	Mouse hepatocytes/hepatoma cell line	A single predicted binding site in the 3´UTR, located at 195 bp	Upregulated	[14]
FOXO1	Forkhead box protein O1	A transcription factor regulating insulin signaling pathway, and played roles in adipogenesis, gluconeogenesis and glycogenolysis	hBMSCs/hFOB1.19 human osteoblast cell line	n.d.	Downregulated	[15]
GDF5	Growth/ differentiation factor 5	A member of the TGF-β superfamily and closely related to the bone morphogenetic proteins (BMPs)	hBMSCs/hFOB1.19 human osteoblast cell line	n.d.	Downregulated	[15]
OCLN	Occludin	Major component of tight junction	MSCs/ATII cells	n.d.	Downregulated	[16]
CK18	Cytokeratin 18	Component of cytoskeleton intermediate filaments	MSCs/ATII cells	n.d.	Downregulated	[16]
PICK1	Protein interacting with C kinase 1	Regulation of traffic between surface receptors	Human breast cancer cell lines	Targeting the 3′-UTR	Downregulated	[17]
IGF2	Insulin-like growth factor 2	Growth factor has growth-regulating, insulin-like and mitogenic activities	MKN28, MKN45, SGC7901 and GES-1 cell lines; NSCLC cell lines.	Directly binds to the 3´-UTR	Downregulated	[18,19]
hTERT	Telomerase reverse transcriptase	A catalytic subunit of the enzyme telomerase	56 different NCI-60 cell lines	Targeting its 3′UTR	Downregulated	[20]
CELF2	CUGBP Elav-like family member 2	A tumor suppressor RNA-binding protein implicated in the regulation of several post-transcriptional events	MKN28, MKN45, SGC7901 and GES-1 cell lines	n.d.	Downregulated	[11]
HMGB3	High mobility group box 3	Multifunctional protein with various roles in different cellular compartments	Human normal bronchial epithelial cell line 16HBE/NSCLC cell lines (A549 and H1299)	n.d.	Downregulated	[21]
JunB	JunB proto-oncogene, AP-1 transcription factor subunit	Transcription factor involved in regulating gene activity following the primary growth factor response	Human ESCC cell lines (Eca-109, TE-1 and KY-SE) and human normal esophageal cell line	Downregulated by NORAD	Upregulated	[22]
AP-1	Transcription factor subunit	Involved in several cellular processes (cell growth, differentiation and apoptosis)	Human ESCC cell lines (Eca-109, TE-1, and KY-SE) and Human normal esophageal cell line	Downregulated by NORAD	Upregulated	[22]
Ywhag	14-3-3-δ, protein kinase C inhibitor protein 1	An abundant, cytosolic and brain-specific protein, which mediates signal transduction	Mice cerebral anterior cortex		Downregulated	[23]
LINGO-1	LRR and Ig domain containing NOGO receptor interacting protein 1	Transmembrane protein selectively expressed in neurons and oligodendrocytes in CNS and the spinal cord, mediating axon growth	NSCs fetal brain 14th E.D.	Bind with the target sites (GGACCCC) in the 3′-UTR located in 202-223bp	Downregulated	[24]

n.d., not defined; PPARδ, peroxisome proliferator-activated receptor gamma; TGF-β, transforming growth factor beta; CNS, central nervous system; NSC, neural stem cells.

**Table 2 cells-09-01566-t002:** Targets directly downregulated by miR-615-5p in cancer.

Target Gene	Description	Gene Action or Effect	Model	Status in Cancer Cells	3′-UTR Targeting Sequence	Effect	Ref
IGF2	Insulin-like growth factor 2	Growth factor	NSLCC, ESCC, HCC, PAC	Upregulated	5’GGACCCCA3’	Good prognosis (clinically); ↓ cell motility, migration, cell proliferation and tumor growth	[9,27,56,57,58]
AKT1 AKT2	Serine/threonine protein kinase B 1 and 2	Regulation of metabolism, apoptosis, cell cycle and transcription	LUAD, PAC	Upregulated	5’GACCCCA3’ 5’GACCCCU3’	↓ tumor growth and metastasis in vivo and cell proliferation, migration and invasion in vitro. ↑ apoptosis	[56,59,60]
SHMT2	Serine Hydroxymethyltransferase 2	Cellular energy metabolism, proliferation and migration	NSLCC, HCC	Upregulated	5’GGACCCC3’	↓ proliferation, migration, and prevented growth of HCC cells	[51,56]
IGFR1	Insulin-like growth factor type 1 receptor	Receptor tyrosine kinase	HCC	Downregulated	5’GGACCC3’	Tumor suppressor effect; ↓ downstream mediators like mTOR	[31]
DDR2	Discoidin Domain Receptor Tyrosine Kinase 2	Receptor tyrosine kinase	T-cell lympho-blastic lymphoma	Upregulated	5’GACCCCAA3’	↑ apoptosis; ↓ cell viability	[54]
EGFR	Epidermal growth factor receptor	Receptor tyrosine kinase	Glioblastoma	Upregulated	5’CCACGAGC3’	Good prognosis (clinically); ↓ cell growth, migration and invasion	[55]
NF-kB2	Nuclear factor NF-kappa-B p100 subunit	Transcription factor related to immunity, differentiation, cell growth, tumorigenesis and apoptosis	Ovarian cancer	Upregulated	5’GGACCCC3’	↓ viability, cell migration and invasion; ↑ apoptosis	[53]
MEF2C	Myocyte-specific enhancer factor 2C	Transcription factor, role in myogenesis, neurogenesis and vasculogenesis	NSLCC	Upregulated	n.d.	↓ cell proliferation, survival, tumor growth, migration and invasion	[61]
JUNB	JunB Proto-Oncogene	Transcription factor, AP-1 transcription factor subunit	PAC	Upregulated	n.d.	↓ cell motility, migration and cell proliferation ↓ HRas/Raf/MAPK and PI3/Akt cascades; ↓ AKT and ERK phosphorylation	[9]
CCND2	Cyclin D2	Cell cycle regulator	Prostate cancer	Upregulated	5’GGACCCC3’	↓ proliferation, migration and invasion of cancer cells in vitro and in vivo	[62]
SF3B3	Splicing Factor 3b Subunit 3	Forms small nuclear ribonucleoproteins complex	NSLCC	Upregulated	5’GACCCC3’	↓ cell proliferation, survival, tumor growth, migration and invasion	[61]
RAB24	Ras-related protein 24	Cytoskeletal remodeling, motility and adhesion	HCC	Upregulated	5’GGACCCC3’	↓ EMT process promotion (↑ E-cadherin, ↓ vimentin, ICAM and β-integrin); ↓ proliferation, survival, motility, adhesion and angiogenesis in vitro and in vivo	[63]
FIBRO-NECTIN-1		ECM protein	LUAD	Upregulated	5′GUGGACCCC3′	↓ MMP2 and MMP9; ↓ migratory and invasive capability	[52]

ECM, extracellular matrix; EMT, epithelial mesenchymal transition; ESCC; esophageal squamous cell carcinoma; HCC, hepatocellular carcinoma; LUAD, lung adenocarcinoma; NF- κB, nuclear factor kappa B; n.d., not defined; NSLCC, non-small cell lung carcinoma, PAC, pancreatic adenocarcinoma; RTK, receptor tyrosine kinase.

**Table 3 cells-09-01566-t003:** Inhibitory molecules of miR-615 3p and 5p.

Molecule	Description	Effect	Cell Line	Status in Cancer Cells	Axis Documented	Effects Associated with Inhibition of miR-615	Ref.
circ-CAMK2A	Circular RNA	Sponge miR-615-5p	LUAD cell lines, HBE	Upregulated	circ-CAMK2A/miR-615-5p/fibronectin-1/MMP	Metastasis, advanced TNM stage and poor prognosis; ↑ migration, and invasion	[52]
circPUM1	Circular RNA	Sponge miR-615-5p	Ovarian cancer cell lines and human peritoneal mesothelial cell line	Upregulated	circPUM1/miR-615-5p/NF-κB	Associated with FIGO stage (poor prognosis); ↑ proliferation, survival, migration, tumor growth and metastasis	[53]
circ-LAMP1	Circular RNA	Sponge miR-615-5p	T-LBL cells Jurkat, CCRF-CEM and SUP-T1	Upregulated	circ-LAMP1/miR-615-5p/DDR2	↑ cell proliferation and viability; ↓ apoptosis	[54]
circRNA-100146	Circular RNA	Sponge miR-615-5p and 3p	16HBE, LUAD cell line	Upregulated	circRNA- 100146/ miR-615-5p and 3p/MEF2C and SF3B3	Poor clinical prognosis; ↑ cell proliferation, survival, tumor growth, migration and invasion	[61]
LINC00324	Long noncoding RNA	Sponge miR-615-5p	LUAD cell lines and 16HBE	Upregulated	LINC00324/ miR-615-5p/AKT1	↑ cell proliferation, migration and invasion ↓ apoptosis	[60]
lncRNA Gm15290	Long noncoding RNA	Sponge miR-615-5p	HBE and NSCLC cell lines	Upregulated	Gm1529/ miR-615-5p/AKT2, IGF2 and SHMT2	↑ proliferation and invasion; ↓ apoptosis	[56]
HOTTIP	HOXA transcript at the distal tip	Sponge miR-615-3p	16HBE/NSCLC cell lines. Human RCC cell lines, normal renal epithelial cells HK-2, 293T.	Upregulated	HOTTIP/miR-615-3p/HMBG3 HOTTIP/miR615-3p/IGF2	Endogenous sponge; promotion of glycolysis under hypoxic conditions in LUAD; lead to suppression of IGF-2 in RCC	[19,21]
NORAD (LIN-C00657)	Noncoding RNA Activated by DNA Damage	Sponge miR-615-3p	Human ESCC cell lines and HEEC	Upregulated	NORAD/miR-615-3p/JunB	Upregulated after DNA damage Oncogenic	[22]
KDM4B	Histone demethylase	Lysine demethylase	HCC cell lines	Downregulated	KDM4/miR-615-5p/RAB24	Demethylation of the miR-615-5p promoter; ↑ proliferation, motility, adhesion and angiogenesis	[63]
CDX2	Caudal type homeobox 2	Transcriptional activator	Pancreatic adenocarcinoma	Downregulated	CDX2/ miR-615-5p/IGF2	Induction of transcription of miR-615-5p; neoplastic cell growth.	[57]
PU.1	Transcription factor	Transcriptional activator	HCC cell lines (Hep3B, MHCC97L and MHCC97H).	Downregulated (in metastatic HCC)	PU.1/ miR-615-5p/IGF2	Induction of transcription of miR-615-5p; ↑ migration and invasion	[64]

FIGO, International Federation of Gynecology and Obstetrics; HCC, hepatocellular carcinoma; LUAD, lung adenocarcinoma; NF-κB, nuclear factor kappa B; NSLCC, non-small cell lung carcinoma; RCC, renal cell carcinoma; TNM, tumor nodes metastasis.

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
