# Peer review of "miR-615 Fine-Tunes Growth and Development and Has a Role in Cancer and in Neural Repair"

_cells, 2020, doi:10.3390/cells9071566_

Round 1

Reviewer 1 Report

The role of miR-615 is well presented and described in this review.

my comments are the following:

  1. the Authors start the paper presenting the two mature transcripts of the miR-615 hairpin but then some function is attributed to miR-615 and not to one of its mature sequence. for instance, Table 1. I suggest the Authors to specify which mature mirna is responsible for the functions showed in that table. Further, it appears that all miR-615-5p are missing from that table (such as VEGF, AKT, etc).
  2.  The role of HOX genes in brain tumors is emerging and I think it should be addressed in this review, also including mir-615 levels in glioma.
  3.  lines 177-197: unclear, please rephrase
  4.  figure 2: it is confusing. the upper and bottom panels are not interconnected so I suggest to separate the two parts of the figure and describe them separately in the legend.

Author Response

Reviewer #1

  1. the Authors start the paper presenting the two mature transcripts of the miR-615 hairpin but then some function is attributed to miR-615 and not to one of its mature sequence. for instance, Table 1. I suggest the Authors to specify which mature mirna is responsible for the functions showed in that table. Further, it appears that all miR-615-5p are missing from that table (such as VEGF, AKT, etc).

RESPONSE: We thank the reviewer for his/her comment and suggestion. We would like to note that throughout the manuscript, in all cases the corresponding miRNA mature form is indicated, whether 3p or 5p, except when, from the revised reference, the original authors did not specify to which form they referred. In the Tables we clearly identified the form to which we refer. In fact, one of the tables is solely for the 3p form (Table 1), another for the 5p form (Table 2) and just one is for both forms (Table 3).

  1. The role of HOX genes in brain tumors is emerging and I think it should be addressed in this review, also including mir-615 levels in glioma.

RESPONSE: We completely agree with the reviewer's comment in terms of the recently studied participation of HOX genes in brain tumors, but believe it is beyond the scope of this review. Instead, we present the involvement of miRNA-615, independently of the HOX cluster in which it is embedded. However, in attention to the interesting reviewer comment, we include the following very recent reference: “Among them, the Hox genes are very well described as specifying the anteroposterior pattern, as well as the development of many organs [2], as for example during neurodevelopment as well as in malignant glial tumors, as gliomas and glioblastomas [reviewed in Gonçalves et al., 2020].”

  1. 177-197: unclear, please rephrase

RESPONSE: We thank the reviewer for bringing to our attention this partially inconsistent paragraph. We rewrite it as follows: “The oncogenic function of miR-615-3p in gastric cancer has been reported recently [10]. In gastric cancer cells (cell line SGC7901), overexpression of miR-615-3p promotes proliferation and migration by suppressing CELF2 (tumor suppressor CUGBP- and ETR-3-like family 2) expression. More recently, elevated miR-615-3p expression has also been implicated in poor prognosis of postoperative biochemical recurrence (PBR) and prostate cancer (PC) specific survival. Since it increases the viability, proliferation, and migration of PC3M prostate cancer cells [11], and is associated with a worse prognosis in PC, miR-615-3p is indicated as an oncogenic driver of this cancer. Moreover, in breast cancer tissues and breast cancer cell lines, miR-615-3p level is also upregulated, and promotes metastatic ability by targeting 3´-UTR of PICK1, inhibiting it, and thus increasing TGF-b signaling. By contrast, PICK1 expression exerts the opposite effect, acting as a negative feedback loop for TGF-b signaling by inhibiting the binding of DICER1 to Smad2/3 and the processing of pre-miR-615-3p to mature miR-615-3p [46].”

  1. figure 2: it is confusing. the upper and bottom panels are not interconnected so I suggest to separate the two parts of the figure and describe them separately in the legend.

RESPONSE: We thank the reviewer for pointing this out. We agree, and have split the figure, generating new Figures 2 and 3. Each has a new explanation in the Figure legend, as follows:

Figure 2. Schematic representation of the effects of MiR-615-5p on cancer. The molecules that are a direct target of miR-615-5p are indicated in blue, and the signaling pathways that are inhibited by the increase in miR-615-5p concentration are schematized, as well as the functional consequences that this implies, which together generate a clear tumor suppressor effect. See the text for further explanation.

Figure 3. miR-615-5p can be negatively regulated under three different paradigms, according to the evidence available at present. It is possible to block the action of the molecule by ceRNAs, either circRNAs or lncRNAs. It is also susceptible to repression by hypermethylation of its intragenic promoter, which has been related to deficiencies in the action of demethylases such as KDM4. Finally, the expression of this miRNA is positively and independently regulated by transcription factors, which, when repressed, cease to induce the expression of miR-615-5p. The functional consequence of this inhibition is the abolition of its tumor suppressor effect, which finally leads to the oncogenic effect associated with the decrease in miR-615-5p concentration.

Reviewer 2 Report

The title of the manuscript sounds well, but it does not cover exactly the content. It is a review from the cited articles we expect to get an exact concrete mechanism or at least a theory for fine tuning, what the authors generated from the previously published data. There is a short section (sect. 8.) about neurological aspects (miR-615 Involvement in Neural Plasticity), why did you put it to the title? Why did you include it?

Summary: I recommend to rewrite it. "The pleiad of functions in which this miRNA is involved, as well as their multiple target genes and the multiple regulatory molecules involved in its own expression, makes it an interesting study case, in which we seek to present in an organized way new knowledge of the mechanisms of epigenetic regulation of miRNAs that may help to define novel therapeutic targets for their associated diseases." It says nothing, while sounds good, but it is empty, as does not have any concrete relation with the content of the ms. We would like to read it, but we do not receive it from this work.

Introduction: Please correct the size of the miRNAs, it is not only 21-22 nt. Later you tell that the size of miR-625 is 96 nt. It is shown in the Figure 1 too, plese explain it.

Table 1. shows interacting genes with miR-625, please make the Effect column wider, explain suppress, depress.

What does it mean physiological and pathological targets?

Section 4. has the title "Mir-165-3p Dual Role as a Tumor Suppressor or Promoter in Cancer", may be oncogene fits better.

Section 5. with the title of "MiR-615-5p as a Tumor Suppressor in Several Types of Cancer". You started to discuss it at Section 4, why did you make a different section?

Table 2., please make this table more clear, it is hard to follow, make wider separations.

Section 7. with the title of "Other miR-615-5p Repressor Mechanisms Which Contribute to the Promotion of Tumor Growth". I would use this section at the end of the ms.

Would be interesting to read about the therapeutical possibilities, how can we interact in different type of cancer. How can you explain the different role of miR-625 during the embyogenesis and cancer development?

I recommend to reorganize the ms and make it more readable and interesting, get conclusions from the cited articles, build up your theory.

Please clarify why did you choose miR-625? There is no microRNA which acts alone, would be interesting to see the networking and where is the place of this miRNA in the network of miRNAs, lncRNAs and circRNAs, may be proteins.

Please use the accepted formula for miRNAs.

Author Response

Reviewer #2

 The title of the manuscript sounds well, but it does not cover exactly the content. It is a review from the cited articles we expect to get an exact concrete mechanism or at least a theory for fine tuning, what the authors generated from the previously published data. There is a short section (sect. 8.) about neurological aspects (miR-615 Involvement in Neural Plasticity), why did you put it to the title? Why did you include it?

RESPONSE: We think that we understand, and share, the reviewer's point in the general sense. At this stage of the knowledge, any expectation of revealing an exact concrete mechanism of action for a miRNA is somewhat unrealistic. Nevertheless, in close attention to the reviewer's comment, we have now split Figure 2 into Figures 2 and 3. Each one includes our view of the molecular mechanisms involved, both in miR-615 expression and in its participation in cancer through its target genes. To the best of our knowledge this is the first time that these two explanations have been presented in literature as an integral view of miR-615 fine-tune regulation.

With regards to the second part of the comment referring to MS section 8, this section was included because when we looked for all the information about miRNA-615, we found that it is not only related with cancer, but also with neural repair and we decided we could not simply omit this fact. This is a hugely different function, from its being oncogenic or acting as a tumor suppressor, and that is why we also settled this in the title, to highlight this totally different aspect.

  1. Summary: I recommend to rewrite it. "The pleiad of functions in which this miRNA is involved, as well as their multiple target genes and the multiple regulatory molecules involved in its own expression, makes it an interesting study case, in which we seek to present in an organized way new knowledge of the mechanisms of epigenetic regulation of miRNAs that may help to define novel therapeutic targets for their associated diseases." It says nothing, while sounds good, but it is empty, as does not have any concrete relation with the content of the ms. We would like to read it, but we do not receive it from this work.

RESPONSE: We agree with the reviewer that some parts of the indicated phrase could be subjective opinions and that the last part of the phrase is mostly speculative, therefore we have rewritten the whole sentence, suppressing our personal opinions, and making it more concise, as follows:

“In addition, we present the pleiad of functions in which miR-615 is involved, as well as their multiple target genes and the multiple regulatory molecules involved in its own expression, by introducing in a comprehensible way the reported knowledge of their actions and interactions and proposing an integral view of its regulatory mechanisms.”

  1. Introduction: Please correct the size of the miRNAs, it is not only 21-22 nt. Later you tell that the size of miR-625 is 96 nt. It is shown in the Figure 1 too, plese explain it.

RESPONSE: This apparent incongruity is related to the nature of the miRNA molecules. The whole miRNA sequence corresponds to the 96 nt, however the portions that interact correspond with the 20-21 nt sections. In fact, any miRNA can include the 3´ and the 5´ sections, of around 20-21 nt within the larger sequence. This is clearly depicted in Figure 1.

  1. Table 1. shows interacting genes with miR-625, please make the Effect column wider, explain suppress, depress.

RESPONSE: Regarding the format of the tables, we would note that the journal will adjust column width and font size to provide the best possible view. However, in this new version we make an extra effort to present the Tables more clearly in the vertical version that the journal generates for reviewing. Regarding the terms used, we simply include the terms used by the authors, which reports the specific miRNA action. Generally, “suppression” means a large depleting or an almost complete elimination, and “depress or decrease” indicates a partial or incomplete depletion.

  1. What does it mean physiological and pathological targets?

RESPONSE: We thank the reviewer for bringing this point to our attention, although the use of these terms depends on the context. By “physiological”, we refer to the characteristic of or appropriate to an organism's healthy or normal functioning, and “pathological” indicates an altered state or one caused by disease. In agreement with the reviewer comment, we rewrite the subtitle 3 as follows: “miR-615 Targets Under Physiological and Pathological Conditions”.

  1. Section 4. has the title "Mir-165-3p Dual Role as a Tumor Suppressor or Promoter in Cancer", may be oncogene fits better

RESPONSE: In agreement with the reviewer comment, we change the subtitle as follows: "miR-615-3p Dual Role as Oncogene and Tumor Suppressor in Cancer"

  1. Section 5. with the title of "MiR-615-5p as a Tumor Suppressor in Several Types of Cancer". You started to discuss it at Section 4, why did you make a different section?

RESPONSE: This was because we separated the participation of the 3p form and the 5p form of miRNA-615, in order to present and discuss them separately. Both forms act in different ways, as we present in this review.

  1. Table 2., please make this table more clear, it is hard to follow, make wider separations.

RESPONSE: We apologize to the reviewer on this point. We sent the editorial office a much better formatted table in horizontal form, but they put it in vertical format, so the spacing was affected. This will change in the final formatted form, but it depends entirely on the journal. Nevertheless, in attention to the reviewer's comments, we have made some adjustments to the table text to favor a better visualization.

  1. Section 7. with the title of "Other miR-615-5p Repressor Mechanisms Which Contribute to the Promotion of Tumor Growth". I would use this section at the end of the ms.

RESPONSE: We thank the reviewer for this suggestion, and while we considered it carefully, this section is in fact a continuation of the previous three sections. All of them focus on the role of miR-615 in cancer. Following it, we still have a final section on the role of this miRNA under a totally different paradigm, that of neural repair. This has very recently been investigated, and we cannot simply obviate or omit it in this review.

  1. Would be interesting to read about the therapeutical possibilities, how can we interact in different type of cancer. How can you explain the different role of miR-625 during the embyogenesis and cancer development?

RESPONSE: We concur with the reviewer that an exploration of the therapeutic possibilities of some interventions, such as CRISPR-Cas and many others, and their interaction with different types of cancer will be of great interest. However, the evidence is still scarce, and it falls beyond the scope of the present review. Regarding the second part of the comment, it is widely recognized that a large number of genes involved in early development are also involved in cancer, and vice versa. This is precisely why, in the present review, in addition to the participation of this particular miRNA (miR-615) in cancer, we also collect, present and analyze the still limited evidence on its role during development. We hope that further research can be undertaken supported by the information presented here.

With attention to the reviewer's interesting comment, we decided to add a final comment as “Future Challenges”, as follows:

“The versatility in the action of miRNAs as powerful regulators of gene expression makes them a therapeutic tool with great potential for the treatment of diverse diseases, including cancer and neurodegenerative diseases (Gareev et al, 2020; Titze-de-Almeida et al, 2020). In this regard, the most studied strategies are: 1) the restitution of the concentration of a repressed miRNA, thus restoring its inhibitory effects on other molecules; 2) the inhibition of an overexpressed miRNA that is pathologically inhibiting some signaling pathway. However, there are still many challenges to be solved, among them how to deliver the miRNAs to the cell of interest, or the control of side effects in the context of a systemic application. Recent work has addressed this issue extensively (Gareev et al, 2020; Christopher et al, 2016). So far, there is no experimental evidence of manipulation of miR-615 for therapeutic purposes.”

  1. Please clarify why did you choose miR-625? There is no microRNA which acts alone, would be interesting to see the networking and where is the place of this miRNA in the network of miRNAs, lncRNAs and circRNAs, may be proteins.

RESPONSE: We have been interested of late in glioblastomas and in the search for biomarkers for their categorization. Therefore, when looking for information, this particular miRNA among all the others attracted our attention for several reasons. We found much apparently conflicting information regarding its actions and effects on cancer, as oncogene or as tumor suppressor, as well as a scarcity of information on its physiological role during development, and finally some information regarding its actions in neural repair. We wanted to compile all this information, to organize and contrast it as reference for future research. In Figures 2 and 3, we present the knowledge relationships of this miRNA and its regulators and demonstrated target genes (not theoretical).

  1. Please use the accepted formula for miRNAs.

RESPONSE: In agreement with the reviewer request, the nomenclature for miRNA has been uniformized throughout the text.

Finally, we would like to thank both reviewers for their detailed revision of our manuscript and for their constructive and highly appreciated comments.

Round 2

Reviewer 2 Report

The authors reflected for most of my questions and comments.

There are a few things which are still problematic.

Title: I still have problem with the title of the ms, I recommended to change it. How do you explain it has dual role in neural repair?

Introduction: We are not agreement in the size of the microRNAs, it is not 21-22 bp, it is rather 18-25 bp.

Table 2, I still do not understand the difference between "suppress" and "depress", up- or down regulated?

Section 8. It is involved in the development of Huntington disease and Alzheimer, how do you explain it, there is no citation related that, just stated without explanation. These two diseases are also in the focus of interest, would be great to see how?

Author Response

Reviewer #2

  1. Title: I still have problem with the title of the ms, I recommended to change it. How do you explain it has dual role in neural repair?

RESPONSE: Regarding this apparently semantic confusion, in the title we refer to miR-615 as “dual role”, because it acts not only in cancer, but also in neural repair. A different aspect would be that in cancer this miRNA can act as an oncogene or as tumor suppressor (a different dual role), however, this aspect is mentioned in the abstract and clearly expounded throughout the manuscript. Nevertheless, attending the reviewer comment, we have removed the term “dual” from the title.

  1. Introduction: We are not agreement in the size of the microRNAs, it is not 21-22 bp, it is rather 18-25 bp.

RESPONSE: Attending to the reviewer amendment, we change the values to 18-25 nucleotides (nt), (not base pairs, because it is a single stranded RNA, although in its secondary structure it performs loops, however the “pb” does not count here).

  1. Table 2, I still do not understand the difference between "suppress" and "depress", up- or down regulated?

RESPONSE: Frankly speaking, this question pertains to the authors of the works that we reviewed here, because in Table 1 (not Table 2), we only reflected on the effect that they reported. Therefore, we leave those terms unchanged to accurately maintain those used by the authors: these effects are not our interpretation. On the other hand, with regards to “suppress” and “depress”, both terms are clearly referring to a down-regulation of the target gene. In Table 2, the situation is slightly different, because there the “status in cancer cells” is regarding the level of expression of the target gene being up or down-regulated. However, accepting that if this point is not totally clear to the reviewer it could also be confusing for the readers, and with an eye to making them more straightforward, we have changed the terms in Table 1 to “up-regulated” or “down-regulated” respectively.

  1. Section 8. It is involved in the development of Huntington disease and Alzheimer, how do you explain it, there is no citation related that, just stated without explanation. These two diseases are also in the focus of interest, would be great to see how?

RESPONSE: We thank the reviewer for this precision. Because this miRNA has been reported as deregulated in both pathologies, and given that the references were commented on previously, we have rewritten this sentence as follows, mentioning again both references. As the reviewer points out, it will of course be of great interest to understand more precisely how this miRNA is involved in neural pathologies.

“Moreover, miR-615 seems to also be involved, to a certain degree, in neurodegenerative diseases such as Huntington’s [85] and Alzheimer´s [86], and most likely in others. Further research is therefore warranted”.